# Effects of Moisture Migration and Changes in Gluten Network Structure during Hot Air Drying on Quality Characteristics of Instant Dough Sheets

**DOI:** 10.3390/foods13193171

**Published:** 2024-10-06

**Authors:** Yuwen Wang, Jie Chen, Fei Xu, Yuqi Xue, Lei Wang

**Affiliations:** 1College of Food Science and Technology, Henan University of Technology, Zhengzhou 450001, China; 13605683485@163.com (Y.W.); xu.fei@haut.edu.cn (F.X.); xuq135699@163.com (Y.X.); wlei0904@163.com (L.W.); 2Henan Province Wheat-Flour Staple Food Engineering Technology Research Center, Zhengzhou 450001, China; 3Henan Province Zhongyuan Food Laboratory, Luohe 462000, China

**Keywords:** instant dough sheets, water migration, gluten network structure, quality

## Abstract

The impact of hot air drying temperature on instant dough sheets’ qualities was investigated based on water migration and gluten network structure changes. The results revealed that the drying process redistributed the hydrogen proton, with deeply bound water accounting for more than 90%. The T_2_ value decreased as the drying temperature increased, effectively restricting moisture mobility. Meanwhile, microstructural analysis indicated that instant dough sheets presented porous structures, which significantly reduced the rehydration time of instant dough sheets (*p* < 0.05). In addition, elevated drying temperatures contributed to the cross-linking of proteins, as evidenced by increased GMP and disulfide bond content (reaching a maximum at 80 °C), which improved the texture and cooking properties. Hence, the water mobility was effectively reduced by controlling the drying temperature. The temperature had a facilitating impact on promoting the aggregation of the gluten network structure, which improved the quality of the instant dough sheets.

## 1. Introduction

Dough sheets are a traditional staple food originating from central China, known for their authentic regional characteristics. Compared to noodles, they have a wide rectangular shape and a thickness of 0.6 to 0.8 mm. In recent years, as people’s pace of life has accelerated, convenient and healthy instant wheat products have gained popularity among consumers. Hence, the combination of traditional dough sheets and modern processing techniques have given rise to the instant dough sheet (IDS). However, currently produced IDSs suffer from issues such as long rehydration times, a lack of chewiness, and poor texture quality. Importantly, it has been found that the process parameters during different processes are the key factors affecting the final quality of the product. Therefore, the study and optimization of the IDS process is crucial. In addition, we have previously studied the IDS steaming process and optimized the optimum parameters (steamed for 6 min) [1]. Based on this, the optimization of the most suitable drying process is necessary, which contributes to the improvement of IDS quality at the source of the process.

The drying process is one of the crucial steps in the production of IDSs, which plays a significant role in ensuring the final product quality. The drying process has a considerable impact on quality properties such as cooking quality and mechanical damage to the noodles [2,3]. Currently, researchers are primarily focused on how the drying process affects the quality of refined dried noodles, instant noodles, and pasta [4,5,6]. However, research data available on the IDS drying process are lacking. In addition to considering productivity and energy consumption, drying parameters should be optimized to maintain maximum product quality. However, the drying process of IDSs is currently designed in an empirical way through trial-and-error runs and based on the practical knowledge of noodle producers rather than the processing theory of industrialization. In addition, it was shown that the diffusion and migration of water, as well as the formation and refinement of the gluten network during the drying process, directly affected the drying characteristics and quality of the noodles during the rehydration process [7,8,9,10]. Nevertheless, during IDS drying, the effects of moisture migration, redistribution, and gluten network structure changes are not clear. Furthermore, the impact of these changes on the final quality of IDSs remains poorly studied. Therefore, in this sense, investigating the variations in the structural properties of IDSs during the drying process is essential for improving product quality. In addition, the design of the drying process is crucial for quality control and quality assurance in IDSs based on the enhancement of product quality.

Accordingly, the aim of this study was to investigate the moisture migration patterns and changes in the structural properties of the gluten network during the drying process of IDSs. In addition, this study also uncovered the impact of structural properties on textural properties and cooking quality. This study provides a theoretical basis for the processing of IDSs and improve their overall quality.

## 2. Material and Methods

### 2.1. Materials and Chemicals

The wheat flour was obtained from Jinyuan Food Co., Ltd. (Zhengzhou, China). The moisture, protein, and lipid contents were 13.69%, 12.2%, and 1.2% for the wheat flour, which were determined according to the AACC (2010). All reagents used in this study were of analytical grade and were provided by Xinfeng Laboratory Co., Ltd. (Zhengzhou, China).

### 2.2. Preparation of Instant Dough Sheets

Figure 1 illustrates the production flow of IDS. The selection of the steaming process parameters referred to Wang et al. [1].

### 2.3. Drying Experiments of IDS

The steamed dough sheets were evenly placed in a stainless steel box and dried at hot air drying temperatures of 60, 70, 80, and 90 °C to a safe moisture content of 10–12% [11]. The drying air velocity was 10 m/s.

### 2.4. Measurement of Drying Characteristics

#### 2.4.1. Determination of Moisture Content

Referring to the direct drying method described in GB 5009.3-2016 for determination [12], the samples were dried in a constant-temperature drying oven at 105 °C until the weight of the samples was constant. The moisture content is calculated according to Equation (1):(1)$$W=\frac{W1-W2}{W1-W3}$$
where *W* denotes the moisture content of IDS, %; *W*1 denotes the weight of the aluminum box and IDS, g; *W*2 denotes the weight of the aluminum box and IDS after drying, g; and *W*3 denotes the weight of the aluminum box, g.

#### 2.4.2. Determination of Moisture Content and Drying Rate of Dry Substrates

The dry samples were measured at 10 min intervals, and the calculations are shown in Equations (2) and (3), respectively:(2)$$Mt=\frac{mt-m}{m}$$
(3)$$DR=\frac{Mt2-Mt1}{t2-t1}$$

In Formula (2), *Mt* is the dry base moisture content of the material at the moment of drying *t*, g/g; *mt* is the mass of the material at the drying time *t*, g; and *m* is the mass of the material at the time of reaching the absolute dry state, g.

In Formula (3), *DR* represents the drying rate, g/(g/min); *Mt*2 represents the dry base moisture content of the material at the moment of drying *t*2, g/g; and *Mt*1 is the dry base moisture content of the material at the moment of drying *t*1, g/g.

### 2.5. Nuclear Magnetic Resonance (NMR) Test

A low-field nuclear magnetic resonance (NMR) analyzer (Suzhou Niumai Analytical Instrument Co., Ltd., Suzhou, China) was used to determine the T_2_ spectra of the IDSs. The test parameters were set to 2500 ms sample interval time, 0.20 ms echo time, 3000 echoes, and 32 scans.

The key parameters of magnetic resonance imaging (MRI) were as follows: repetition time = 500 ms, TE = 20 ms, and RG = 20 db.

### 2.6. Chemical Bonds and Intermolecular Interactions

#### 2.6.1. Measurement of Disulfide Bonds and Free Sulfhydryl (SHfree) Content

The content of disulfide bonds and free sulfhydryl groups (SHfree) was determined according to Peng et al. [13] with minor modifications: the samples (240 mg) were dissolved in 4 mL of buffer A (0.2 mol/L Tris-HCl, 8 mol/L urea, 3 mM EDTA, 1% SDS, and pH 8.0), mixed for 1 h, and centrifuged at 8000× *g* for 20 min.

Determination of free sulfhydryl groups: 0.1 mL buffer B (10 mM DTNB in 0.2 M Tris-HCl, pH 8.0) was added in 4 mL of the supernatant. Take the supernatant (2 mL) after centrifugation as described above and measure the absorbance at 412 nm (T9P, AOE Instruments Co., Ltd., Shanghai, China). Tris-Gly buffer was used as a blank control. SH-free content was calculated by Equation (4).

Determination of the total sulfhydryl: a 5 mg sample was reduced with dithiothreitol (40 mM DTT in 0.2 M Tris-HCl, pH = 8.0, 1 mL) at 60 °C for 2 h. The total sulfhydryl contents were determined by 5,5′-dithiobis-2,2′-nitrobenzoic acid (DTNB) method. Finally, the absorbance was measured at 412 nm. The disulfide bond content was calculated by Equation (5).
(4)$$SHfree\;(\mu \mathrm{mol}/g)=\frac{A412\times D}{a\times b\times C}$$
(5)$$S-S\;(\mu \mathrm{mol}/g)=\frac{N2-N1}{2}$$

In these equations, *a* = 1.36 × 104; *b*, the cell path length; *A*412, the absorbance of the samples at 412 nm; *D* = 1.04; *C*, the concentration of samples, mg/mL; *N*1, the free thiol group content; *N*2, the total sulfhydryl content.

#### 2.6.2. Measurement of Non-Covalent Bonding Interactions

The non-covalent bonding interaction content was determined according to Yang et al. [14]. A standard curve was drawn with bovine serum protein (Sigma, Darmstadt, Germany) to determine the protein content (mg/g) in solutions.

### 2.7. Glutenin Macropolymer (GMP) Content

The GMP content in the samples was measured according to Song et al. [15]. The protein content was determined by the Kjeldahl nitrogen (Kjeltec 8400, Foss, Rockwall, TX, USA), just as the GMP content was measured.

### 2.8. SDS-PAGE

SDS-PAGE analyses (reducing and non-reducing conditions) were performed according to Si et al. [16] with minor modifications. Some 5% (*v*/*v*) β-mercaptoethanol (β-ME) was added to the reducing condition. The gel was stained with 0.25% (*w*/*v*) Coomassie Brilliant Blue G-250 and then destained with 10% (*v*/*v*) acetic acid to visualize the protein bands.

### 2.9. Protein Secondary Structure

The protein secondary structures were analyzed by a Fourier transform infrared spectroscopy (FTIR) spectrometer (IRAffinity-1S, Simadzu Corporation, Kyoto, Japan). A mixed sample (25~30 mg) (potassium bromide: sample = 100:1) was taken and pressed for 2 min for measuring.

### 2.10. Microstructure

The microstructure of IDS was determined according to Wang et al. [1].

### 2.11. Rehydration and Cooking Properties

Rehydration and cooking properties were determined according to the AACC method 66–50 (AACC, 2010) and Wang et al. [17]. The relevant calculations are shown in Equations (6) and (7):(6)$$Water\;\mathrm{absorption}\;\mathrm{rate}\left(\%\right)=\frac{(M2-M1)}{M1}\times 100\%$$
(7)$$Cooking\;\mathrm{loos}\left(\%\right)=\frac{(M4-M3)\text{×}10}{M1}\times 100\%$$

In these equations, M1, 10 g of sample; M2, weight of samples reaching rehydration time; M3, the weight of the solution that reached the rehydration time; M4, the solution was dried at 105 C to a constant weight.

### 2.12. Measurement of Gelatinization Degree

The gelatinization was determined using the approach of Wang et al. [18]. The degree of gelatinization is calculated in Equation (8):(8)$$Gelatinization\;\mathrm{degree}\;(\%)=\frac{\left(Y-P3)-(Y-P4)-(Y-Q\right)}{\left(Y-P1)-(Y-P2)-(Y-Q\right)}\;\times \;100\%$$

In the formula, *Y* is the consumption of sodium thiosulfate in distilled water, and P1, P2, P3, P4, and *Q* are the amounts of sodium thiosulfate consumed in bottles A1, A2, A3, A4, and B, respectively.

### 2.13. Texture Properties

Texture properties were determined and slightly modified by referring to Javaid et al. [19]. The texture properties of the samples (IDS for achieving rehydration time) was performed using a texture analyzer (TA-XT Plus, Stable Micro Systems, Godalming, UK). The TPA test parameters were as follows: HDP/PFS probe, 1 mm/s pre- and post-test speed, 0.8 mm/s test speed, 65% strain displacement.

### 2.14. Statistical Analysis

The data acquired in the paper were expressed as “mean ± standard deviation (SD)”. Significant differences between the data were analyzed using the Duncan test in SPSS 21.0 software, which were considered significant at *p* < 0.05. Origin 2021 software (Origin Lab, Inc., Northampton, MA, USA) was used to plot figures.

## 3. Results and Discussion

### 3.1. Water Distribution and Migration

#### 3.1.1. Drying Characteristics of IDS

Drying is the process of moisture dissipation. The drying characteristic curve typically describes the trend in the moisture content of the material over time. The moisture content of the IDS decreased with drying time (Figure 2A). Meanwhile, the moisture loss rate of IDS was faster at the initial stage than at the final stage. The time to reach a safe moisture content was 40 min less for drying at 90 °C than for drying at 60 °C (≤10%). Thus, the moisture content was reduced significantly by raising the drying temperature. This is due to the fact that rising drying temperatures are associated with a greater temperature difference in the heat transfer driving force [20]. Nevertheless, in terms of the quality of IDSs, especially the external appearance, drying over 80 °C was not optimal (Figure 2C).

Commonly, the drying rate of food materials consists of a rising drying rate period, a constant drying rate period, and a falling drying rate period [21]. Figure 2B displays the drying rate curve of IDSs. The drying rate of the IDSs increases with increasing temperature; the IDS dried at 60 °C exhibited the minimal drying rate. Thus, increasing the drying temperature significantly reduces the drying time. At the beginning stage of drying, the sample is preheated and warmed up, and the drying rate increases from zero to reach the maximum value, which belongs to the accelerated drying stage. Subsequently, the drying rate shows a decreasing trend into the decelerated drying stage. The overall trend of IDSs was in the rising–falling drying rate period, with no clear constant drying rate period emerging. This is due to a fact that a thin layer of the sample (0.7 mm) could not contribute to a constant moisture supply during the drying process. As a result, the rate at which moisture migrates from the interior to the exterior of the IDS is lower than the rate at which moisture evaporates from the exterior. This results in an unsaturated moisture vapor pressure on the ample surface [22]. This confirmed that the drying temperature was excessively high and that the product exhibited the bubbling phenomenon (Figure 2C). These results revealed that internal moisture migration was the primary element governing moisture transfer during IDS drying, and that the drying rate depends on the rate of internal moisture diffusion [23].

#### 3.1.2. Water State of IDSs during Drying Process

The state, distribution, and migration of water affect the formation of the gluten network and the processing properties of IDSs. The T_2_ spectra of the IDSs at various drying time points are plotted in Figure 3A–D. The sample exists in the form of T_21_ (deeply bound water), T_22_ (weakly bound water), and T_23_ (free water) during the drying process, with T_21_ predominating. In addition, the mobility of water reduces with decreasing transverse relaxation time (T_2_), which implies an increase in the ability of water to bind to non-aqueous substances. The T_21_ showed a decreasing trend as drying time or temperature increased (Figure 3A–D). This phenomenon illustrated that the water protons may be exchanged with starch hydroxyl protons during drying that were gradually removed from the intragranular space. Conversely, the water protons came into contact with CH protons in amorphous starch and gluten networks, thus restricting the mobility of the water [8,24,25].

Table 1 displays the proportions of the water states during the drying process of the IDSs. The results demonstrate that the water that existed in the dried IDS was primarily deeply bound water (A_21_ > 90%), a significant increase of 60% compared to the control, whereas the weakly bound water decreased significantly (A_22_: 0.6–4.78%) (*p* < 0.05). These outcomes further verified that the drying process potentiated the binding capacity of the remaining water interior of the IDS, making it difficult to evaporate in the middle and final drying periods (30–60 min). However, during the drying process, the mobility of partial water presented a tendency to increased, which manifested as a rise in the free water percentage (A_23_: 0.96–7.92%). During the initial drying phase, the water protons are in close contact with the gluten network or the internal space of the starch granules, resulting in the formation of a porous structure within the IDS in which vapor can be present and transferred (Figure 4) [26].

#### 3.1.3. Water Distribution and Migration of IDS during Drying Process

The water migration and redistribution in the IDSs during drying were measured by nuclear magnetic resonance imaging (MRI). In proton imaging, dark blue indicates a background without water, and the color transition from green to yellow to red represents an increase in proton density [10,27].

The proton density image of the IDS is shown in Figure 3E. Before drying (0 min), the IDS evenly distributed water, exhibiting a high proton density (red region). As drying proceeded, the red region on the image diminished from the edges of the IDS towards the center, indicating that evaporation was initiated from the edges of the IDS, with moisture in the center evaporating afterwards. This was consistent with the drying results of Chinese-dried noodles [4]. Furthermore, the red region remained after drying at 60 °C for 15 min, but after drying at 90 °C for 5 min, the red region in the center basically disappeared. It can be concluded that increasing the drying temperature could accelerate the water evaporation of the samples and significantly shorten the drying time, which was consistent with the aforementioned results.

### 3.2. Microstructure

In order to visualize the effect of drying on the quality attributes of the IDSs, the microstructure of samples was investigated using scanning electron microscopy (SEM). The microstructure of the IDSs is shown in Figure 4. SEM captured the fact that the IDS drying process led to the formation of widely distributed pores. The pore structure of the IDSs facilitated water diffusion during rehydration by providing a spatial structure for water to penetrate into the interior of the sample. This is attributed to the rapid heating of the food during the drying process, dehydration, evaporation, and immediate pressurization. When the internal vapor pressure reaches or exceeds the structural tension, it promotes the overall expansion of the sample and the formation of a loose and uniform microporous structure [28]. However, dense structures were displayed within the low-temperature dried samples (Figure 4A,B), indicating the presence of a small number of large pores (red squares). By elevating the drying temperature (Figure 4C), the interior of the sample displayed a multiple and continuous macroporous structure (red circle). These changes in internal porosity explained why the IDSs dried at different temperatures had different rehydration properties. This means that drying at high temperatures improves the rehydration properties. However, the starch granules swelled slightly under high drying temperatures (90 °C) and destroyed the surface protein layer [3,29]. Moreover, this expansion restricts the migration of moisture within the sample, which reduces the drying rate. Thus, cracks (red arrows) began to appear inside the IDS (Figure 4D). Hence, an appropriate drying temperature is critical to optimizing the quality of IDSs.

### 3.3. Structural Changes in the Gluten Network

#### 3.3.1. Chemical Bonds and Intermolecular Interactions

The gluten network structure is established by the interaction of gluten protein molecules through intermolecular covalent (disulfide) and non-covalent (hydrogen bond and hydrophobic interactions) bonds. According to Table 2, the free sulfhydryl group content remarkably decreased as the temperature increased from 60 to 90 °C, revealing a reciprocal relationship with the variation of disulfide bond content. A previous study demonstrated that during the drying process, the protein unfolded, resulting in the oxidation of free SH groups and the conversion to disulfide bonds. This conversion was related to the aggregation of gluten [30]. This could be confirmed by the increased GMP content (Figure 5A). Hence, elevating the drying temperature facilitates the formation of the gluten network by increasing the proportion of SS/SH exchange, especially for the IDS with a drying temperature of 80 °C. However, drying at 90 °C, the disulfide bond content started to decrease, and it could be surmised that the aggregation of the gluten network reaches equilibrium at a certain temperature and that exceeding this temperature destroys the structural integrity of the gluten network [5,31].

Correspondingly, non-covalent interactions also varied with temperature. After the drying process, the hydrogen bond content exhibited an upward and then downward trend, reaching a maximum at 80 °C. The level of intermolecular interaction implies a high degree of cross-linking of gluten proteins [32]. Therefore, an appropriate increase in drying temperature could strengthen the intermolecular forces between gluten molecules, forming a dense and strong gluten network. Nonetheless, hydrophobic interactions tend to increase after 80 °C; it is possible that excessive drying temperatures resulting in the denaturation and stretching of proteins expose hydrophobic amino acid residues [33].

The GMP content reflects the degree of gluten cross-linking and aggregation. Figure 5A displays the changes in the GMP content of the IDSs during drying. Consequently, raising the drying temperature (60–80 °C) induced the polymerization of alcohol-soluble proteins from the monomeric structure via sulfhydryl–disulfide exchange, which increased the disulfide bond content [34]. This resulted in an upward trend in GMP content, which was confirmed through increased disulfide bonding content. However, protein structure was negatively affected at 90 °C drying, resulting in a significant decrease in GMP content. High drying temperatures cause damage to the gluten network, since they leads to changes in starch–protein and starch–starch interactions [31].

#### 3.3.2. SDS-PAGE

SDS-PAGE was used to further observe the changes in protein subunits. Figure 5B,C show the SDS-PAGE electrophoretic profile of IDS proteins during the drying process. The number of electrophoretic bands did not change in different samples, indicating that the protein subunit composition remained intact during the drying process. However, the intensity of the electrophoretic bands changed significantly. The electrophoretic spectra under non-reducing conditions (Figure 5B) showed that the intensity of the bands in parts a and b at 60–80 °C followed a weaker trend. Especially after 60 °C, the bands in part b almost disappeared. These markedly attenuated protein band intensities suggested that high-temperature drying induced the covalent cross-linking of disulfide bonds between gluten networks [35].

The electrophoretic spectra of the IDSs under reducing conditions (Figure 5C) are significantly different from those of Figure 5B, as evidenced by an increase in the number of electrophoretic bands and a deepening of the color. This was due to the fact that β-ME broke the disulfide bonds in the protein gel structure and admitted more low-molecular-weight proteins into the strips [36]. The intensity of the bands in parts a and c was significantly strengthened when the drying temperature was increased from 60 to 80 °C. At this stage, the results of the disulfide bonds and GMP content further demonstrate that drying promotes the cross-linking of proteins for aggregation via disulfide bonds. However, when the drying temperature reached 90 °C, a clear weakening of the a and b parts of the bands could be clearly seen in the reducing spectra, which might be caused by the destruction of the glutenin network structure by the excessively high drying temperature. This corresponded to the decreased content of disulfide bonds.

#### 3.3.3. Protein Secondary Structure

FTIR analyzes the secondary structure of proteins in wheat flour on the basis of changes in the amide I band (the secondary structures of gluten protein: 1600–1700 cm^−1^). The different absorption wave numbers of the amide I band correspond to the α-helix (1652–1665 cm^−1^), β-turn (1665–1682 cm^−1^), β-sheet (1610–1640 cm^−1^), and random coil (1640–1652 cm^−1^) in the secondary structure of the protein, respectively.

In Figure 5D, the secondary structure of proteins during the drying process of the IDS was dominated by β-sheet (34.75–36.62%) and β-turn (26.74–30.29%), followed by α-helix (17.11–18.79%) and random coils (16.56–18.75%). In general, the β-sheet and α-helix are regarded as the most stable and ordered protein secondary structures, which act synergistically with intermolecular disulfide bonds to stabilize the gluten network structure [37]. The proportion of β-sheet increased to 36.62% with increasing drying temperature, revealing that this process enhanced the stability of the protein secondary structure. An increase in β-sheet content after heat treatment of wheat flour was also found in a report by Ma et al. [38]. In addition, the α-helix and β-sheet contain a large number of hydrogen bonds, providing a certain degree of rigidity to the secondary structure of proteins. This was relevant to the increased hardness and chewiness in IDS texture properties (Table 3). However, the α-helix began to decrease at 90 °C, indicating the dissociation of proteins and rearrangement of wheat gluten molecules at excessively high temperatures [39].

### 3.4. Changes in the Quality of the IDS

#### 3.4.1. Rehydration and Cooking Properties

The quality of an IDS is affected by rehydration time and cooking losses. The quality of an IDS can be evaluated by the rehydration time, which is usually considered to be less than 360 s; otherwise, it is considered to be less instant [5]. During the drying process, the high temperature and the water in the IDS cause the starch to undergo gelatinization, i.e., α-degree. In general, increasing the α-degree of starch can effectively enhance the rehydration of the IDS. According to Figure 6, the rehydration time of the IDS dried at 80–90 °C (273 s) was observed to be comparable but significantly shorter than that at 60 °C (362 s), with a significant increase in the degree of gelatinization to 89.55%, suggesting that increasing the drying temperature facilitated the gelatinization degree of starch and promoted the rehydration process.

Cooking loss is an important indicator of flour product quality. Generally, products with low cooking losses are more acceptable to consumers. As displayed in Figure 6, the cooking loss of the IDSs showed a downward trend and then an upward trend as the drying temperature increased, reaching a minimum at 80 °C. Moreover, an optimal drying temperature could boost the cooking quality of the samples. This was attributed to the cross-linking and aggregation of proteins during the drying process, creating a more stable gluten network structure [40]. However, the decrease in cooking quality of IDSs at 90 °C drying implied that drying over 80 °C disrupted the gluten network structure, thereby increasing cooking losses.

#### 3.4.2. Texture Properties

Water diffused and penetrated from the surface to the interior of the IDS during rehydration, influencing its texture properties. Generally, good-quality IDSs have the property of chewiness. Table 3 illustrates the textural parameters of IDSs. The results demonstrated that raising the drying temperature significantly increased the hardness and chewiness of IDSs (*p* < 0.05). However, dry temperature had a minimal impact on IDS springiness. Based on the aforementioned findings, it was evident that high temperatures could promote the cross-linking and aggregation of proteins, resulting in the formation of a dense and stable gluten network structure. This increased the hardness and chewiness of the IDSs, resulting in a high-quality product. Interestingly, the IDS gluten network structure showed a tendency to weaken when dried at 90 °C. However, the hardness of the IDS continued to increase. This is due to the continuous breaking of hydrogen bonds between IDS gluten proteins and starch during drying at 90 °C, prompting starch gelatinization along with protein coagulation [41]. In addition, IDS dried at 60–80 °C tended to increase adhesiveness, which could be attributed to the increased gelatinization of starch with increasing drying temperature (Figure 6).

### 3.5. Pearson’s Correlation Analysis

In order to further elucidate the relationship between the moisture distribution, gluten network structure, and starch properties of IDS and quality, the results of correlation analysis are shown in Figure 7. The rehydration time of the IDSs was significantly and negatively correlated (*p* < 0.05) with the hot air drying temperature and the gelatinization degree of starch. The results indicated that the hot air drying temperature had a significant effect on the pore structure formed inside the IDS and the gelatinization properties of starch, which significantly affected the rehydration properties of the IDS. The internal porosity and starch gelatinization degree of the IDS were increased by increasing the drying temperature, which significantly reduced the rehydration time of the IDS and facilitated the rapid rehydration of the product. The hardness of the IDSs was significantly correlated with α-helix, β-sheet, and free sulfhydryl group (*p* < 0.05), and its chewiness was significantly correlated with drying temperature, weakly bound water, β-sheet, and free sulfhydryl group (*p* < 0.05). This suggests that protein secondary structure, water distribution, and disulfide bonds have a significant effect on the hardness and chewiness of IDSs. The drying process enhances the binding capacity of water and non-aqueous substances (mainly proteins and starch) within an IDS, thus reducing water migration. In addition, drying promotes the cross-linking of proteins to form a dense and stable gluten network structure. This effectively increases the hardness and chewiness of IDSs.

## 4. Conclusions

In conclusion, the drying process could improve the quality of an IDS by altering its structural properties. The findings of this study revealed that hot air drying of IDSs occurred only in the rising and falling rate periods, and the magnitude of the drying rate was dependent on the drying temperature. Furthermore, the drying process redistributes hydrogen protons, resulting in a significant increase in the proportion of deeply bound water (A_21_ > 90%). This change facilitated the tight binding of water and non-aqueous substances (protein and starch), reducing water mobility. SEM revealed that the IDS was porous, with numerous and small porosities formed during water diffusion and evaporation inside, facilitating rehydration rates and rapid rehydration. Additionally, increasing the drying temperature enhanced the stability of the gluten structure by promoting the cross-linking and aggregation of proteins, which was reflected in the increased disulfide bonding, GMP content, and β-sheet ratio. This resulted in a firm, chewy IDS—a high-quality product. However, a comprehensive analysis illustrated that a moderate drying temperature (80 °C) would promote the formation of a dense gluten network, responsible for the excellent rehydration attributes, texture, and cooking qualities. The research data in this paper and our previous study [1] demonstrated that steaming and hot air drying was effective in improving the quality of IDSs. These process findings provide theoretical support for future industrial applications.

## Figures and Tables

**Figure 1 foods-13-03171-f001:**
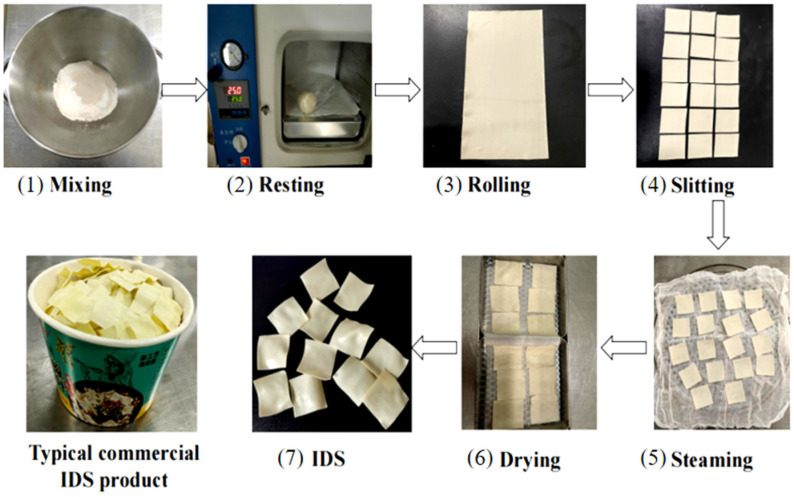
Flow chart of IDS production. (1) Mixing for 6 min (100 g of wheat flour, 38 g of water). (2) Resting for 20 min at 25 °C. (3) Sheeting into thin sheets (0.7 mm). (4) Slitting into dough sheets (4 × 4 cm). (5) Steaming at 121 °C, 0.4 MPa for 6 min. (6) Drying in hot air dryer. (7) Getting instant dough sheets.

**Figure 2 foods-13-03171-f002:**
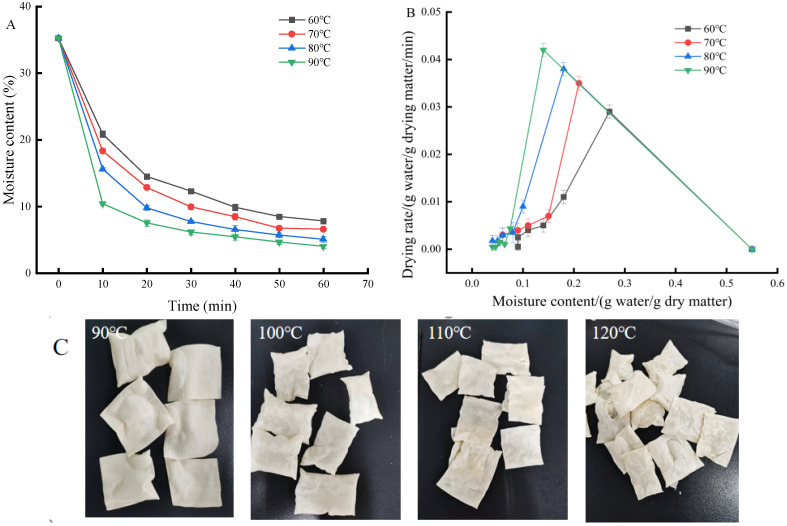
Hot air drying characteristics of IDSs. (**A**) Moisture content variation curves of IDSs under different hot air drying temperatures; (**B**) drying rate variation curves of IDSs under different hot air drying temperatures; (**C**) finished IDS products after hot air drying at different temperatures.

**Figure 3 foods-13-03171-f003:**
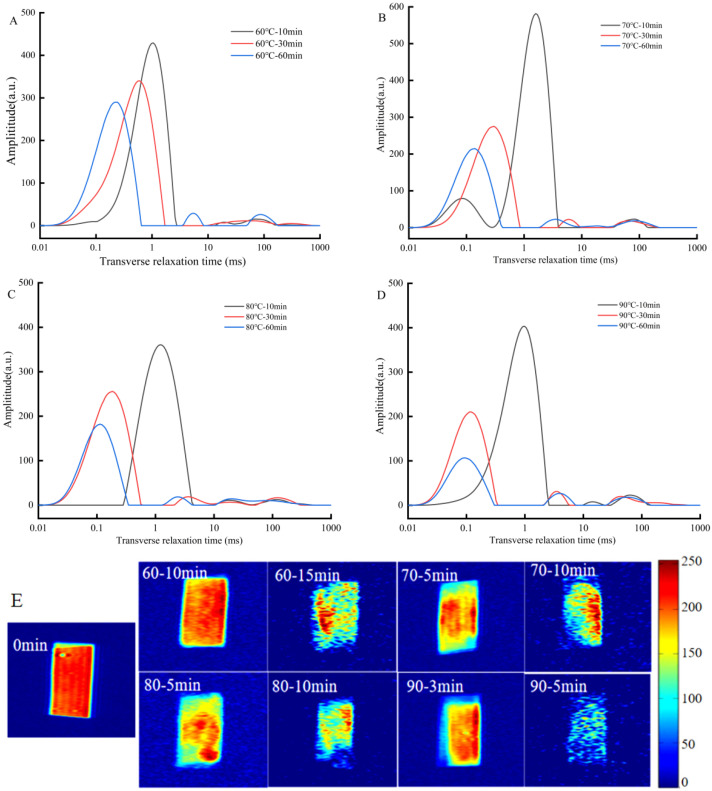
Moisture distribution and migration of the IDSs during the drying processes. (**A**–**D**) are the transverse relaxation time T2 spectra during the drying process at 60, 70, 80, and 90 °C, respectively. (**E**) MRI images of IDSs at different drying temperatures; 60-10 min means drying at 60 °C for 10 min; the other explanations in (**E**) are likewise. The dark blue color in the figure represents the anhydrous background and the blue-to-yellow-to-red color represents the increased hydrogen proton density, indicating the increased water content.

**Figure 4 foods-13-03171-f004:**
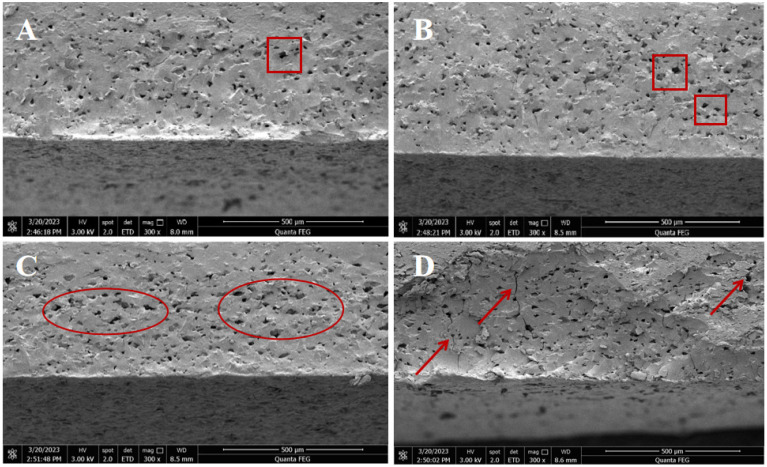
Microstructure of IDS at different hot air drying temperatures. (**A**–**D**) are SEM images dried at 60, 70, 80, and 90 °C, respectively. Red circles represent continuous macroporous structure; red squares represent a small number of large pores; red arrows represent cracks.

**Figure 5 foods-13-03171-f005:**
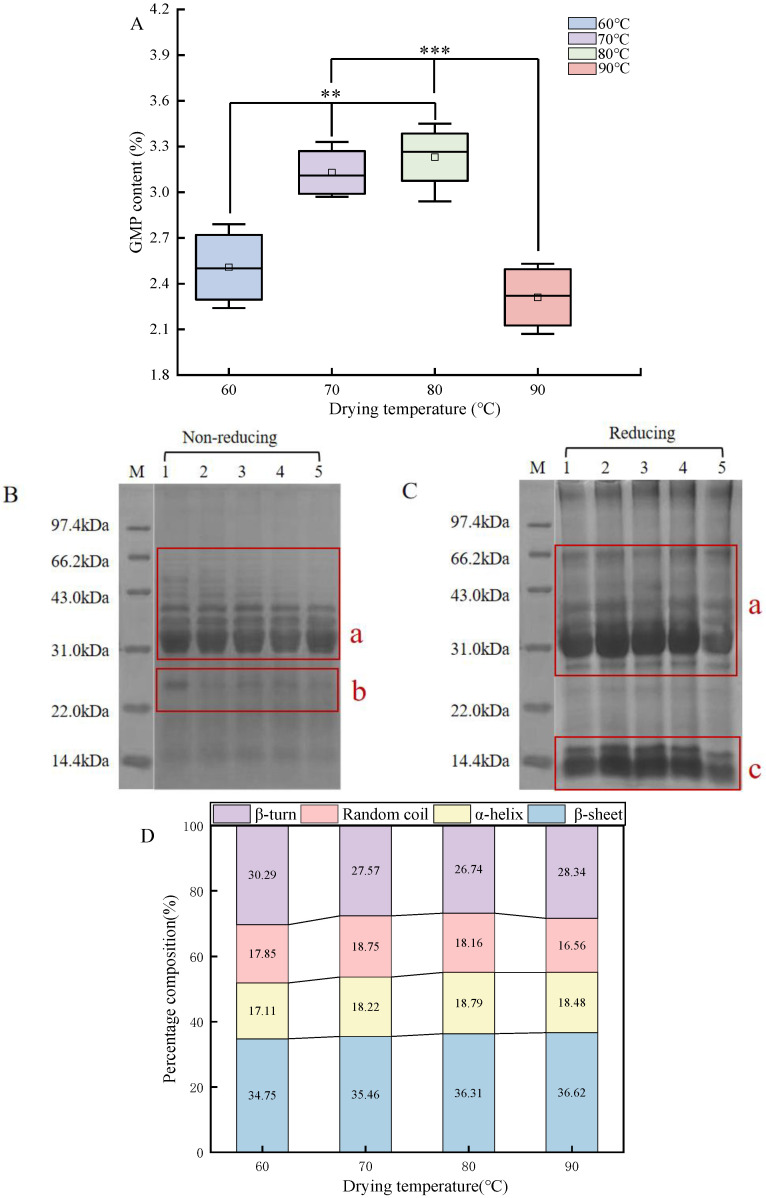
Structural changes in the gluten network of the IDS during drying. Note: (**A**) Changes in GMP content of IDS during the drying process. “**”represents *p* < 0.01; “***” represents *p* < 0.001; (**B**,**C**) SDS-PAGE analysis of samples under non-reducing conditions and reducing conditions. The numbers 1, 2, 3, 4, and 5 correspond to the IDS dried at 0, 60, 70, 80, and 90 °C respectively; (**D**) Protein secondary structure percentage diagram.

**Figure 6 foods-13-03171-f006:**
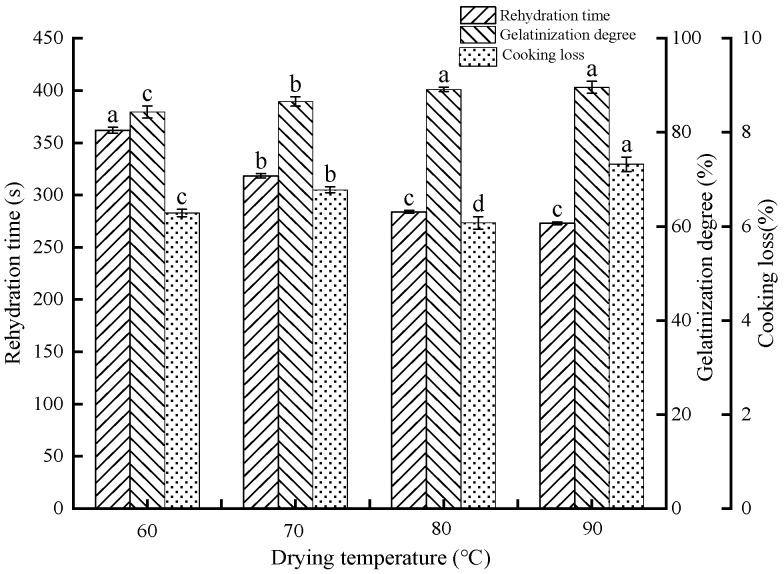
IDS rehydration properties and cooking quality at different drying temperatures. Note: Different letters in the same column demonstrate significant differences (*p* < 0.05).

**Figure 7 foods-13-03171-f007:**
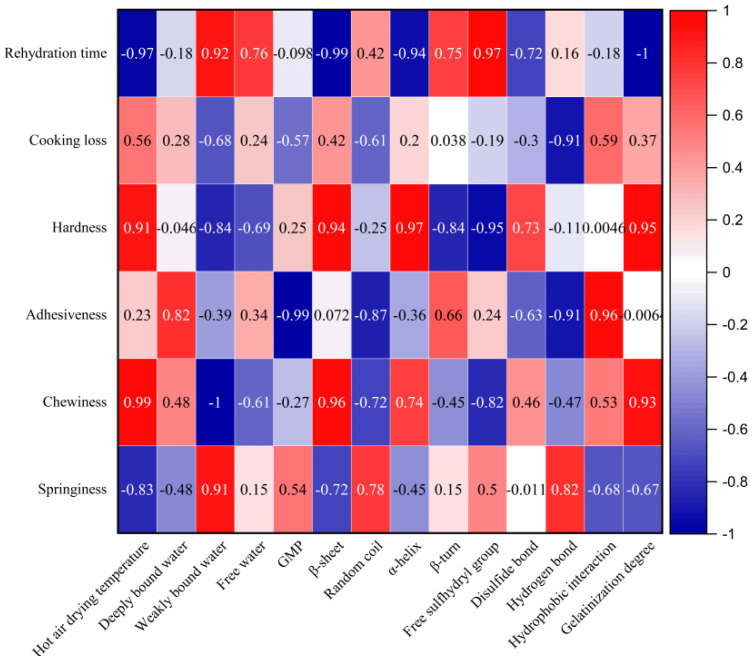
Pearson’s correlation analysis between moisture morphology, gluten network structure, texture properties, and cooking properties of IDS.

**Table 1 foods-13-03171-t001:** Changes in T_2_ corresponding peak area during drying of IDS.

Drying Process	Peak Area (A_2_)		
Dry Temperature-Time (°C-min)	A_21_ (%)	A_22_ (%)	A_23_ (%)
0	34.89 ± 0.68 ^e^	65.00 ± 0.65 ^a^	0.06 ± 0.02 ^f^
60-10	98.31 ± 0.52 ^a^	0.61 ± 0.92 ^e^	0.96 ± 0.06 ^e^
60-30	95.80 ± 0.41 ^b^	2.01 ± 0.47 ^cd^	2.03 ± 0.03 ^d^
60-60	95.98 ± 0.71 ^b^	2.17 ± 0.59 ^c^	3.98 ± 0.02 ^c^
70-10	97.61 ± 0.50 ^a^	0.92 ± 0.53 ^e^	2.29 ± 0.03 ^d^
70-30	95.65 ± 0.72 ^b^	1.74 ± 0.37 ^d^	3.85 ± 0.07 ^c^
70-60	91.98 ± 0.47 ^d^	2.17 ± 0.32 ^c^	5.64 ± 0.03 ^b^
80-10	95.86 ± 0.61 ^b^	1.38 ± 0.66 ^d^	2.05 ± 0.01 ^d^
80-30	94.90 ± 0.51 ^bc^	2.09 ± 0.60 ^cd^	3.67 ± 0.07 ^c^
80-60	90.20 ± 0.37 ^d^	4.78 ± 0.71 ^b^	4.61 ± 0.04 ^b^
90-10	96.09 ± 0.83 ^ab^	0.60 ± 0.92 ^e^	3.05 ± 0.03 ^c^
90-30	93.32 ± 0.68 ^c^	2.80 ± 0.84 ^c^	3.43 ± 0.02 ^c^
90-60	90.02 ± 0.59 ^d^	3.26 ± 0.77 ^bc^	7.92 ± 0.08 ^a^

Note: Different letters in the same column demonstrate significant differences (*p* < 0.05). Abbreviations: The abbreviations in the table indicate different drying times at different temperatures; e.g., “60-10” means drying at 60 °C for 10 min.

**Table 2 foods-13-03171-t002:** Changes in covalent and non-covalent bonds during drying process of IDSs.

Drying Temperature (°C)	Free Sulfhydryl Group (μmol/g)	Disulfide Bond (μmol/g)	Hydrogen Bond (mg/mL)	Hydrophobic Interaction (mg/mL)
60	4.81 ± 0.04 ^a^	2.14 ± 0.02 ^c^	1.34 ± 0.07 ^b^	1.64 ± 0.04 ^b^
70	4.32 ± 0.02 ^a^	2.80 ± 0.05 ^b^	1.36 ± 0.05 ^b^	1.52 ± 0.03 ^c^
80	3.82 ± 0.04 ^b^	3.94 ± 0.04 ^a^	1.59 ± 0.06 ^a^	1.55 ± 0.02 ^c^
90	3.97 ± 0.03 ^b^	2.83 ± 0.03 ^b^	1.05 ± 0.08 ^d^	1.73 ± 0.02 ^a^

Note: Different letters in the same column demonstrate significant differences (*p* < 0.05).

**Table 3 foods-13-03171-t003:** Effect of hot air drying on TPA texture of IDS.

Drying Temperature (°C)	Hardness (g)	Adhesiveness (g.s)	Chewiness	Springiness
60	5477.10 ± 136.75 ^c^	−79.71 ± 45.99 ^b^	3673.38 ± 96.65 ^d^	0.93 ± 0.02 ^a^
70	7476.06 ± 117.60 ^b^	−112.77 ± 30.73 ^a^	4422.54 ± 78.98 ^c^	0.91 ± 0.01 ^a^
80	8063.34 ± 79.44 ^a^	−123.06 ± 29.48 ^ab^	5483.10 ± 114.26 ^b^	0.92 ± 0.01 ^a^
90	8283.24 ± 96.47 ^a^	−58.80 ± 9.96 ^c^	6880.98 ± 13.47 ^a^	0.86 ± 0.00 ^b^

Note: Different letters in the same column demonstrate significant differences (*p* < 0.05).

## Data Availability

The original contributions presented in the study are included in the article, further inquiries can be directed to the corresponding author.

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
