# Peer review of "Effects of Moisture Migration and Changes in Gluten Network Structure during Hot Air Drying on Quality Characteristics of Instant Dough Sheets"

_foods, 2024, doi:10.3390/foods13193171_

Round 1

Reviewer 1 Report

Comments and Suggestions for Authors

Excellent paper. I have only one suggestion:

Please under Figure 1 just describe what is presented on the figure. All description about procedure for production of IDS transfer like as text in subtitle 2.2.

Author Response

Comments 1: Please under Figure 1 just describe what is presented on the figure. All description about procedure for production of IDS transfer like as text in subtitle 2.2.

Response 1: We sincerely thank the reviewer’s comments. We agree with this comments. According to the reviewer’s comments,We have revised the statement as follows according to the reviewer’s comments:

Line 74-77: Figure 1. Flow chart of IDS production. (1) Mixing for 6 minutes (100g of wheat flour, 38g of water). (2) Resting for 20 minutes at 25 °C. (3) Sheeting into thin sheets (0.7-mm). (4) Slitting into dough sheets (4 cm×4 cm). (5) Steaming at 121°C, 0.4 MPa for 6 minutes. (6) Drying in a hot air dryer. (7) Getting instant dough sheets.

Reviewer 2 Report

Comments and Suggestions for Authors

In my opinion, the Authors offer many results, but some of the points made need to be clarified. My specific comments are inserted in the Authors pdf, but some of the concerns I will add here. First of all, the Methods section needs thorough checking since I have noticed a lot of mistakes. Additionally, Authors have published paper with similar subject recently, so the Methods overlap. Maybe consider shortening. Also in the Results and Discussion section, I suggest that Authors combine some sections, for instance, 3.11., 3.1.2. and 3.1.3 since they overlap and at the same time they can harmonize conclusions. Figure 2B needs to be checked since I think it Is not plotted correctly, and hence the discussion around it also needs to be corrected. In my opinion chapter 3.1.1. needs corrections, and as mentioned it would be best to coordinate it with 3.1.2. and 3.1.3. The same applies to the section with GMP and SH groups, but it is mandatory for the section dealing with water distribution and migration. 

Comments on the Quality of English Language

Minor English revision is needed. 

Author Response

Comments 1: In my opinion, the Authors offer many results, but some of the points made need to be clarified. My specific comments are inserted in the Authors pdf, but some of the concerns I will add here. First of all, the Methods section needs thorough checking since I have noticed a lot of mistakes. Additionally, Authors have published paper with similar subject recently, so the Methods overlap. Maybe consider shortening. Also in the Results and Discussion section, I suggest that Authors combine some sections, for instance, 3.11.,3.1.2. and 3.1.3 since they overlap and at the same time they can harmonize conclusions. Figure 2B needs to be checked since I think it Is not plotted correctly, and hence the discussion around it also needs to be corrected. In my opinion chapter 3.1.1. needs corrections, and as mentioned it would be best to coordinate it with 3.1.2. and 3.1.3. The same applies to the section with GMP and SH groups, but it is mandatory for the section dealing with water distribution and migration.

Response 1: Thanks for the reviewer’s comments. According to your proposal, we have amended the text accordingly and marked them in red. For the Methods section of the text we have checked carefully and revised and abbreviated accordingly. Secondly, the content of the manuscript was integrated according to the appropriate nature. Finally, before plotting Fig. 2B, we referred to the relevant references (Wang et al., 2018; Ogawa & Adachi 2017; Zhou et al., 2015; Yu et al., 2018) for the drying curves, and we have also cited the relevant in the manuscript. We sincerely hope that the revised manuscript will be acceptable to you.

References:

Wang, Z., Yu, X., Zhang, Y., Zhang, B., Zhang, M., & Wei, Y. (2019). Effects of gluten and moisture content on water mobility during the drying process for Chinese dried noodles. Drying Technology, 37(6), 759-769.

Ogawa, T., & Adachi, S. (2017). Drying and rehydration of pasta. Drying Technology, 35(16), 1919-1949.

Zhou, M., Xiong, Z., Cai, J., & Xiong, H. (2015). Convective air drying characteristics and qualities of non-fried instant noodles. International journal of food engineering, 11(6), 851-860.

Yu, X., Wang, Z., Zhang, Y., Wadood, S. A., & Wei, Y. (2018). Study on the water state and distribution of Chinese dried noodles during the drying process. Journal of Food Engineering, 233, 81-87.

Reviewer 3 Report

Comments and Suggestions for Authors

The manuscript addresses an interesting and relevant topic that aligns well with the scope of the journal. The methods employed are technically sound in their current form; however, the manuscript would benefit from a more comprehensive explanation of the mechanistic effects influencing the formation of the material's chemical composition throughout the process. This would significantly enhance its scientific value. The results are largely presented in terms of functional aspects, but they fall short of fully elucidating the experiments' underlying nature. To improve clarity, a more detailed discussion of the molecular state of the materials during their transformation is necessary. Additionally, the manuscript lacks any statistical analysis, which is a critical shortcoming. Incorporating a formal statistical analysis, such as multivariate or cause-effect analysis, would greatly elevate the scientific rigor and quality of the work.

Author Response

Comments 1: The manuscript addresses an interesting and relevant topic that aligns well with the scope of the journal. The methods employed are technically sound in their current form; however, the manuscript would benefit from a more comprehensive explanation of the mechanistic effects influencing the formation of the material's chemical composition throughout the process. This would significantly enhance its scientific value. The results are largely presented in terms of functional aspects, but they fall short of fully elucidating the experiments' underlying nature. To improve clarity, a more detailed discussion of the molecular state of the materials during their transformation is necessary. Additionally, the manuscript lacks any statistical analysis, which is a critical shortcoming. Incorporating a formal statistical analysis, such as multivariate or cause-effect analysis, would greatly elevate the scientific rigor and quality of the work.

Response 1: Thanks for the reviewer’s comments. Firstly, the aim of this paper is to investigate the changes in water molecule movement and migration, structural properties of the gluten network and starch properties during the drying process of instant dough sheets, and to further clarify the effects of the structural changes on the functional aspects of the instant dough sheets (mainly the eating quality). Therefore, we also discuss in the paper the mechanism of changes in moisture distribution and migration, gluten network structure, starch and microstructure during the drying process. It was found that the drying process led to a redistribution of hydrogen protons, increasing the proportion of bound water, promoting the formation of GMP and disulphide bonds, and contributing to the stabilization and refinement of the gluten network structure by altering the structure of the protein subunits and the secondary structure of the proteins, which improved the eating quality of the product.

Secondly, according to your suggestion, we have added a correlation analysis at the end of the article (Line 446-464) to further clarify the relevance of moisture, gluten network structure, and starch properties to the quality of the product, and to enhance the completeness of the article.

Line 446-464: In order to further elucidate the relationship between moisture distribution, gluten network structure and starch properties of IDS and quality, the results of correlation analysis are shown in Figure 7. The rehydration time of IDS was significantly and negatively correlated (p < 0.05) with the hot air drying temperature and the gelatinization degree of starch. The results indicated that the hot air drying temperature had a significant effect on the pore structure formed inside IDS and the gelatinization properties of starch, which significantly affected the rehydration properties of IDS. The internal porosity and starch gelatinization degree of IDS were increased by increasing the drying temperature, which significantly reduced the rehydration time of IDS and facilitated the rapid rehydration of the product. The hardness of IDS was significantly correlated with α-helix, β-sheet and free sulfhydryl group (P<0.05), and its chewiness was significantly correlated with drying temperature, weakly bound water, β-sheet and free sulfhydryl group (P<0.05). This suggests that protein secondary structure, water distribution, and disulfide bonds have a significant effect on the hardness and chewiness of IDS. The drying process enhances the binding capacity of water and non-aqueous substances (mainly proteins and starch) within IDS, thus reducing water migration. In addition, drying promotes the cross-linking of proteins to form a dense and stable gluten network structure. This effectively increases the hardness and chewiness of IDS.

Round 2

Reviewer 3 Report

Comments and Suggestions for Authors

The manuscript is of adequate quality, the data is relevant and could be of interest to readers, I did not find  flaws in the methodology, and the discussion section is sufficient and in accordance with the manuscript objectives. In my opinion, the manuscript is ready to be published. 

Author Response

Comments: The manuscript is of adequate quality, the data is relevant and could be of interest to readers, I did not find  flaws in the methodology, and the discussion section is sufficient and in accordance with the manuscript objectives. In my opinion, the manuscript is ready to be published. 

Response 1: We sincerely thank the reviewer for taking the time to review our manuscript and agreeing with our manuscript! If you have any other questions about this manuscript, we will be happy to answer them.
